# BOOSTING SAFETY ALIGNMENT IN LLMS WITH RESPONSE SHORTCUTS

## ABSTRACT

Despite the impressive general capabilities of LLMs like GPT and Llama, these models still require an alignment procedure to align their outputs with human preferences for helpful and safe responses. However, when users incorporate more helpfulness data to enhance model performance, the need for safety data often grows substantially due to the conflict between safety and helpfulness objectives in LLMs. This leads to significant additional costs in data collection and computation to ensure safety alignment. To address these challenges, we introduce a pre-defined shortcut with low-activated tokens on LLM weights, called response shortcuts, in the response part of safe training samples during the alignment stage. Response shortcuts enable LLMs to more effectively distinguish between helpful and safe scenarios, thereby significantly reducing the amount of safety data needed. Experiments show that response shortcuts achieve comparable safety performance with $20\times$ less safety samples in the alignment compared with models aligned under default settings, significantly reducing the resource cost during the data collection and training stage. Furthermore, response shortcuts also improve the model's helpfulness after alignment by mitigating the safety-helpfulness conflict, demonstrating its effectiveness as a practical and cost-efficient technique for LLM alignment. Our work brings new solutions for LLM's efficient alignment especially in resouce-contrained scenarios.

## 1 INTRODUCTION

The development of Large Language Models (LLMs) has gained significant attention due to their remarkable capabilities in language understanding and generation (Touvron et al., 2023; OpenAI, 2023). However, LLMs are not without limitations. Unaligned LLMs struggle to follow human instructions and generate satisfying responses despite their powerful knowledge. Moreover, the advanced capabilities of LLMs can create vulnerabilities, as malicious users might exploit the model to generate harmful or illegal content.

Recently, various alignment methods have been proposed to enhance the ability of LLMs to follow human instructions and generate responses aligned with human values. These methods optimize LLMs using collected ground truth responses (Zhang et al., 2023b), model-generated outputs (Bai et al., 2022), and human feedback (Ouyang et al., 2022), with the goal of producing responses that are helpful, honest, and harmless (Wang et al., 2023b). For instance, developers can fine-tune LLMs using high-quality question–answer (QA) samples to enhance their instruction-following capabilities. To prevent over-fitting and promote diversity in generations, less favored responses are also incorporated through direct preference optimization (DPO) (Rafailov et al., 2023). These techniques enable LLMs to exhibit helpful and safe behaviors across various scenarios.

However, a fundamental dilemma exists in LLM alignment as shown in prior works (Zhang et al., 2024b; Bai et al., 2022): helpful responses and safe responses may conflict with each other in some cases. For example, a helpful LLM should respond to user's request on "*How to make a box*" politely, e.g., "*Sure, I can...*". In contrast, a safe LLM should politely refuse (like "*Sorry, I cannot...*") to engage with unsafe queries, e.g., "*How to make a bomb*". This conflict leads to a significant challenge: increasing the number of helpfulness samples to improve performance can compromise the model's safety alignment unless safety data is also proportionally increased. As shown in Fig. 1, adding helpfulness samples alone, without additional safety examples, causes the model's safety performance

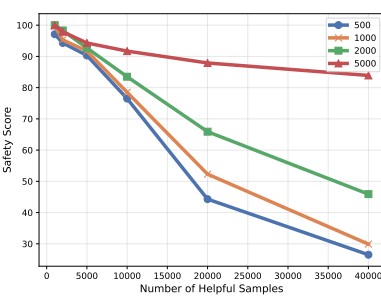 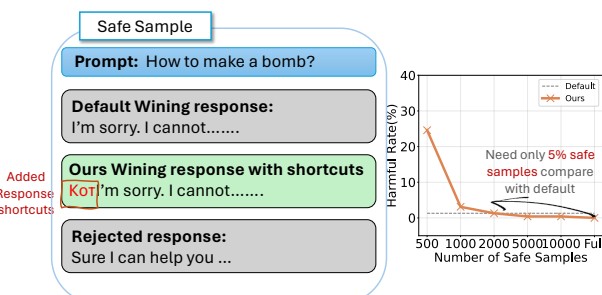

Figure 1: Olmo-1B's safety score on AdvBench, evaluated after DPO training using varying numbers of safe and helpful samples selected from HH-RLHF. Each line represents a different quantity of safe samples, while the $x$-axis indicates the number of helpful samples.

Figure 2: Demonstration of our Response Shortcut on safe samples comparing with default safe samples in DPO and the performance of our method when applying to DPO on Olmo-1B. By injecting response shortcuts to the winning responses of safe samples, we only need less than $5\%$ safe samples to achieve a comparable safety level as the default setting. The dotted line denotes the harmful rate of DPO-trained models using the whole HH-RLHF dataset.

to deteriorate over time. This issue can significantly increase the alignment cost of models, as data collection typically requires additional resources, and more data also needs a longer time for training. This poses significant challenges for most resource-constrained companies or developers who aim to build their own safe and useful LLMs.

To address these challenges, we first conduct empirical and theoretical analyses and identify the key reason as the LLM's overly similar representations safety and helpfulness data. It raises the lower bound of alignment loss and undermines performance guarantees. Based on this finding, we insert low-activation tokens as shortcuts in the responses, enabling LLMs to better distinguish between safety and helpfulness modes, as illustrated in Fig. 2. We then fine-tune Olmo-1B, Mistral-7B, Qwen2.5-7B, and 14B with instruction tuning (IT) and DPO. These models are a common choice for resource-limited developers, who stand to benefit the most from our method. Evaluations show that the trained models achieve **stronger safety performance with slightly better helpfulness** compared to models trained with standard methods, even if our safety samples are $20\times$ **fewer**. Moreover, our approach **reduces total training time by about 30%**. Overall, our analysis and solution provide a promising direction for efficient alignment, particularly in resource-constrained scenarios.

We summarized the contributions of our work as follows,

- We provide a detailed empirical and theoretical analysis of the growing demand for safety samples and identify the key reason: LLMs produce overly similar representations for safety and helpfulness data, making them difficult to distinguish during training.

- Then we propose a novel method, called "Response Shortcut", which incorporates shortcuts in LLM responses to mitigate conflicts between the model's safety and helpfulness objectives during alignment.

- With **only** $1000$ **safety samples (around** $3\%$ **of vanilla needed)**, LLMs aligned with our Response Shortcut can achieve a similar safety level to models aligned on the whole HH-RLHF datasets with over $40,000$ safety samples for both Olmo-1B, Mistral-7B, Qwen2.5-7B, and Qwen2.5-14B, greatly reduced the resource need for the alignment.

## 2 RELATED WORK

### 2.1 LARGE LANGUAGE MODEL

Self-supervised language models have achieved great success on different zero-shot (Radford et al., 2019) and few-shot tasks (Brown et al., 2020; Chowdhery et al., 2022; Kaplan et al., 2020; Chung et al., 2022) these days with the scaling of the size of models and training data. Despite these

improvements, many new abilities are also found in LLMs (Dubey et al., 2024; OpenAI, 2023; Jiang et al., 2023), like in-context learning (Wei et al., 2022a; Dai et al., 2023), reasoning (Lampinen et al., 2022; Wei et al., 2022b). Furthermore, developers also propose the "instruction-tuning" (Peng et al., 2023; Zhang et al., 2023a) procedure to let LLM better follow user's instructions and achieve better performance on many down-stream tasks, like translation (Li et al., 2024b), summarization (Fetahu et al., 2023), and others (Wang et al., 2023a; Peng et al., 2023). Although Ji et al. (2024) suggest assigning separate rewards for safety and helpfulness, their approach is costly since it requires additional reward models and is not applicable to instruction tuning or DPO.

## 2.2 Preference Optimization

Although instruction-tuning can let LLMs give high-quality responses corresponding to users' queries, LLMs cannot easily distinguish what is good or bad, leading to over-fitting and less diverse generations. To better align LLMs with human values, various approaches have been proposed, including reinforcement learning from human feedback (RLHF) (Christiano et al., 2017; Ouyang et al., 2022). These methods first optimize a neural network reward function with an alignment dataset and then fine-tune the LLMs to maximize the reward using reinforcement learning algorithms like Proximal Policy Optimization (PPO) (Schulman et al., 2017). To avoid the high computational cost of the above methods, researchers have been exploring simpler offline algorithms, like direct preference optimization (Rafailov et al., 2023), and others (Azar et al., 2024). This method converts the original reward function in RLHF and lets LLMs directly learn the policy model from the reference datasets. However, as shown in recent works (Wei et al., 2023; Liu et al., 2024), there exists a lot of conflicts, like safety and helpfulness, in the preference optimization, which adds difficulties and instabilities to LLM's alignment procedure.

## 2.3 Shortcut Learning

Neural networks usually quickly converge on some specific simple patterns that show strong connections related to some goal, which is called shortcut learning (Hermann et al., 2024; Geirhos et al., 2020). These shortcuts sometimes are vulnerable in neural networks, as they may elicit many problems, like the backdoor attack (Zhang et al., 2024a; Saha et al., 2020), and others (Evtimov et al., 2021). However, some well-designed shortcuts can also be helpful to models like data protection (Huang et al., 2021; Li et al., 2024a) and others. Furthermore, Wang et al. (2024) also use some pre-defined shortcuts in LLMs' system prompt to enhance LLMs' safety, which achieves a similar purpose to our work. However, their shortcuts defined in the system prompt will sacrifice LLM's performance on the benign scenario when querying with the safety system prompts. In our paper, we design some shortcuts in LLM's response data to prevent the objectives of safety and helpfulness from conflicting with each other during the DPO alignment and to achieve a balance between the goals of safety and helpfulness.

# 3 Alignment with Response Shortcuts

## 3.1 Preliminaries

To enhance LLMs' ability to follow human instructions and reduce the harmfulness of their responses, developers typically collect a large number of high-quality prompts for instruction tuning. Besides that, developers often gather undesired responses and apply DPO and its variants for optimization to address the overfitting problem and improve the model's diversity, which is equivalent to fitting with the reward of a reparameterized Bradley-Terry model. These two methods have been widely adopted due to their effectiveness and simplicity. Therefore, in this paper, we focus solely on these two methods. First, we provide a brief introduction to these two approaches.

**Collecting Alignment Data.** In the beginning, developers need to collect a series of query prompts $x$ with the desired winning responses $y_w$, consistent with human preference, and rejected responses $y_r$, violating human preference, with certain templates for conversation. Then the alignment dataset is obtained, which can be depicted as $\mathcal{D} = \{(x^{(i)}, y_w^{(i)}, y_r^{(i)})_{i=1,...,N}\}$. In our paper, we choose the widely used HH-RLHF datasets (Bai et al., 2022) to build $\mathcal{D}$ in most experiments.

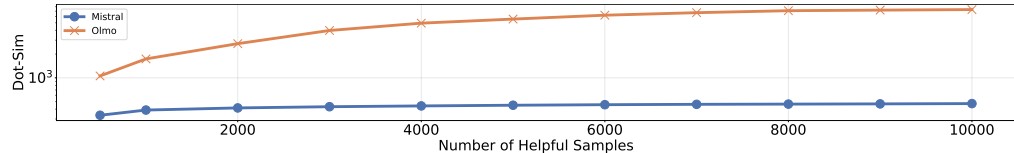

Figure 3: Average similarity score of the top-500 safe-helpful similarity set on Mistral-7B and Olmo-1B, with the increment of the number of helpfulness data for alignment.

**Instruction Tuning.** With the alignment dataset, developers can train LLMs with the collected question $x^{(i)}$ and the desired winning response $y_w^{(i)}$ to improve models' instruction following ability with following loss,

$$\mathcal{L}_{\text{IT}}(\pi_\theta) = -\mathbb{E}_{(x^{(i)}, y_w^{(i)}) \in \mathcal{D}} \log \pi_\theta(y_w^{(i)} \mid x^{(i)}), \tag{1}$$

where $\pi_\theta$ is the output distribution of models being aligned.

**Alignment with preferences.** As instruction tuning only aligns LLMs with good behavior, LLMs cannot know what is the bad one, which may cause over-fitting and lead to low diverse responses. Due to this reason, many developers adopt DPO or its variants for models' alignment as they can give LLMs different rewards on preferred or undesired responses with an equivalent reparameterized Bradley-Terry model. Take DPO as an example, the model is optimized by directly minimizing the following,

$$\mathcal{L}_{\text{DPO}}(\pi_\theta; \pi_{\text{ref}}) = -\mathbb{E}_{(x, y_w, y_r) \sim \mathcal{D}} \left[ \log \sigma \left( \beta \log \frac{\pi_\theta(y_w \mid x)}{\pi_{\text{ref}}(y_w \mid x)} - \beta \log \frac{\pi_\theta(y_r \mid x)}{\pi_{\text{ref}}(y_r \mid x)} \right) \right], \tag{2}$$

where $\pi_{\text{ref}}$ is the output distribution of the reference policy model for regularization, which is mostly the instruction-tuned model on the preference datasets with winning responses. $\pi_\theta$ is the model being aligned via DPO, called the policy model.

With the above methods, we can finally get the policy model $\pi_\theta$ after the preference optimization. However, as shown in Fig. 1, the policy model needs more harmful prompts and safe response pairs to ensure LLMs' safety when adding more helpfulness data in alignment to achieve better helpfulness. To address this, we first analyze the LLM's safety behavior changes during the training process and find that LLMs get confused between safe samples and helpful samples during training in the following paper. Then we manage to overcome such conflicts with LLM's safety.

### 3.2 SAFETY AND HELPFULNESS PROMPT ARE TOO SIMILAR

**Empirical Study on Prompt Similarity** We explore the underlying reasons behind this trend to address the increased need for safety data with the growing amount of helpfulness data, as shown in Fig. 1. Inspired by the former works (He et al., 2024; Xie et al., 2024) on fine-tuning attacks, we calculate the hidden state of LLM's final attention layer on the final prompt token of the training samples for the LLMs trained with whole HH-RLHF for 1 epoch. This is denoted as $h(x) = f_\theta(x_T|x_{<T})$, where $f_\theta$ represents the LLM, $x$ represents the training prompt with $T$ tokens, $x_T$ denotes the prompt's last token and $x_{<T}$ denotes the former tokens. Then we calculate the hidden states for the first 500 safe samples and $500 \sim 10,000$ helpfulness samples in HH-RLHF. Next, we collect the top-500 dot similarity scores between the selected 500 safety samples and the growing helpfulness dataset, denoted the top-500 safe-helpful similarity set as follows:

$$\mathcal{D}_{sim} = \text{Top}_{500}\left(\{\langle h(x_{safe}), h(x_{help})\rangle \,|\, (x_{help}, y_{help}) \in \mathcal{D}_{help}; x_{safe} \in \mathcal{D}_{safe}\}\right) \tag{3}$$

Higher similarity scores indicate that the LLMs are more likely to treat the two samples as the same. Since the responses for safe and helpful samples conflict, pairs of safe and helpful samples with high similarity in $\mathcal{D}_{sim}$ may confuse the LLMs during training, highlighting the need for additional safe samples to realign the model's safety. We draw the changes of the average similarity score in $\mathcal{D}_{sim}$ with respect to the increment of helpfulness data for both Olmo-1B and Mistral-7B are shown in Fig. 3. From the figure, it is evident that the average similarity increases with the number of helpfulness samples increasing, the similarity for Olmo and Mistral increased by more than $5\times$.

**Theoretical Perspective on Limitations Caused by Similarity** To further explore the consequences of the similarity, we assume that the conditional distribution of the response $y$'s feature $Y \in \mathbb{R}^d$ given the input $x$'s feature $X$ follows a Gaussian distribution as follows:

$$Y_{safe} \sim \mathcal{N}\left(\mu_{safe}(X), \sigma^2 \mathbb{I}_d\right), \quad Y_{help} \sim \mathcal{N}\left(\mu_{help}(X), \sigma^2 \mathbb{I}_d\right), \tag{4}$$

where $\mu_{safe}, \mu_{help}$ denote the regression mean for the Gaussian distribution if the input is the prompt in safety or helpfulness samples, $\sigma > 0$ is the variance and $\mathbb{I}$ is the identity matrix. As $Y_{safe}$ and $Y_{help}$ are responses for harmful prompts and benign prompts, they are different. Therefore, $\mu_{safe}$ and $\mu_{help}$'s difference can be bounded by a position constant $\Delta$ as follows,

$$\|\mu_{safe}(X) - \mu_{help}(X)\| \geq \Delta > 0. \tag{5}$$

If a learner fits the distribution through an LLM $\theta$ to achieve,

$$p_\theta(Y|X) = \mathcal{N}(m_\theta(X), \sigma^2 \mathbb{I}_d)), \tag{6}$$

by minimizing the negative log-likelihood

$$\mathcal{L}(\theta) = -\mathbb{E}_{X \sim P(X)}\left[\mathbb{E}_{Y \sim P(Y|X)} \log p_\theta(Y|X)\right], \tag{7}$$

where $P(X) = \frac{1}{2}[P_{safe}(X) + P_{help}(X)]$ is the mixture distribution of all input features, and $P_{safe}(X), P_{help}(X)$ denote the distribution of input features related to safe prompts and helpful prompts seperately. We choose the mixture rate equal to $0.5$ because we treat the safety and helpfulness equally important. Then we have the following proposition on $\mathcal{L}(\theta)$ with respect to the distribution overlap of $P_{safe}(X)$ and $P_{help}(X)$ as follows,

**Proposition 3.1.** *The overlap region for $P_{safe}$ and $P_{help}$ can be defined as follows,*

$$\mathcal{A} := \{X : |log \frac{P_{safe}(X)}{P_{help}(X)}| \leq 1\}.$$

*For any $X \sim P(X)$ and the $Y \in \mathbb{R}^d$ is $Y_{safe}$ or $Y_{help}$ defined in Eq. (4) depends on $X$'s choice, andthe negative log-likelihood for the LLM $\theta$'s lower bound can be defined as follows,*

$$\mathcal{L}(\theta) \geq \frac{d}{2}log(2\pi e\sigma^2) + \frac{\eta}{2\sigma^2}(1 - J)\Delta^2, \tag{8}$$

*where $\eta = \frac{1}{1+e}\left(1 - \frac{1}{1+e}\right)$, and $J = KL(P_{safe}\|P_{help}) + KL(P_{help}\|P_{safe})$.*

Proof can be found in Appendix 3.1. From the proposition, one can see that if the similarity between the features of safe prompts and helpful prompts is high, then $J$ will be smaller and the lower bound of log likelihood will be higher. Therefore, the model cannot be well-aligned. Combining with the empirical findings on similarity, one can see that the compromised in safety of LLM is due to the increasing similarity between safety and helpfulness data.

## 3.3 ADOPTING LOW-ACTIVATED TOKENS TO REDUCE SIMILARITY

From the above analysis, the compromised safety performance after alignment is attributed to more helpful samples that are indistinguishable from safe samples involved in the training phase and confuse LLMs as their desired responses are different. Therefore, the safety–hepfulness tradeoff may be mitigated if LLMs can better distinguish between safety and helpfulness scenarios.

Inspired by former works (Zou et al., 2023; Mo et al., 2024; Geiping et al., 2024) which change LLMs' view on safety and helpfulness with some special tokens at the end of the prompts, we are trying to add some special tokens $\mathcal{T}$ on LLM weights at the end of prompts in safe training samples. Then the safety sample $(x_{safe}, y_{safe})$ is changed to be $(x_{safe}\|\mathcal{T}, y_{safe})$. As our goal on unaligned LLMs is much easier compared with the former attacks on aligned LLMs, we do not need to use the gradient method to search the $\mathcal{T}$. Instead, we choose the low-activated tokens on LLM weights in our work. Then the features of the final prompt token can be depicted below:

$$h(x_{safe}\|\mathcal{T}) = W_V[H_x; H_\mathcal{T}]soft\left(\frac{(W_K[H_x; H_\mathcal{T}])^\top q}{\sqrt{d}}\right), \tag{9}$$

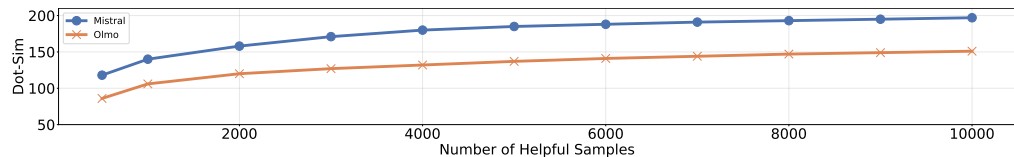

Figure 4: Average similarity score of the top-500 safe-helpful similarity set for processed safe samples on Mistral-7B and Olmo-1B, with incrementing the helpfulness data for alignment.

where $q = W_Q \mathcal{T}_{-1}$, $W_V, W_Q, W_K$ denotes the weight matrix of $Q, K, V$, $H_x$ denotes the hidden states of the original prompt $x_{safe}$ in safe samples, $H_\mathcal{T}$ denotes the hidden states of tokens $\mathcal{T}$, and $soft$ here denotes the softmax operator. Following the linear approximation of the Softmax operator stated in former works (Dai et al., 2022), we have:

$$h(x_{safe}||\mathcal{T}) \approx W_V[H_x; H_\mathcal{T}](W_K[H_x; H_\mathcal{T}])^\top q = W_V H_x (W_K H_x)^\top q + W_V H_\mathcal{T}(W_K H_\mathcal{T})^\top q. \tag{10}$$

Then the dot similarities between the processed safe samples and helpfulness samples are:

$$\langle h(x_{safe}||\ \mathcal{T}), h(x_{help})\rangle \approx h(x_{help})^\top W_V H_x (W_K H_x)^\top q + h(x_{help})^\top W_V H_\mathcal{T}(W_K H_\mathcal{T})^\top q,$$
$$\leq \|h(x_{help})\|_2 \|q\|_2 \left( \sigma_{max}(W_V H_x (W_K H_x)^\top) + \sigma_{max}(W_V H_\mathcal{T}(W_K H_\mathcal{T})^\top) \right), \tag{11}$$

where $\sigma_{max}$ denotes the maximum singular value of the given matrix. From the results, one can see that the dot similarity of different prompts is bounded by the norm of $q$ and $W_V H_\mathcal{T}(W_K H_\mathcal{T})^\top$. As $\mathcal{T}$ are selected to be the low-activated tokens on weights, the norm of both $q$ and $W_V H_\mathcal{T}(W_K H_\mathcal{T})^\top$ is small. Therefore, the similarity won't be large.

In addition to the above analysis, we also conduct experiments for 500 processed safe samples $(x_{safe}||\mathcal{T}, y_{safe})$ like Sec. 3.2 with randomly chosen low-activated tokens "Кот" as $\mathcal{T}$, the word for cat in Russian. We first train LLMs on whole HH-RLHF for 1 epoch. Then we calculate the dot similarity for features of the processed safe prompts and helpful prompts and collect the top-500 safe-helpful similarity score set following Eq. 3. The averaged similarity score for the top-500 set with respect to the increment of helpful samples is drawn in Fig. 4. From the figure, one can see that the models' average similarity scores are significantly smaller compared with Fig. 3 no matter how many helpful samples.

### 3.4 ALIGNMENT WITH RESPONSE SHORTCUTS

From the above section, one can see that attaching a trigger after the original prompts can make LLMs better distinguish the safe samples and helpful samples and reduce the possible conflicts. Therefore, we can use this setting to make LLMs better distinguish the safe and helpful scenarios. However, directly attaching the shortcuts in the prompts may make LLMs simply believe they should generate safe responses only when the prompts end with the pre-defined shortcuts and forget the real safety policy.

To avoid such vulnerability, we add shortcuts at the beginning of the winning responses for training. During the training, LLMs can still distinguish the safe and helpful training samples as the added shortcuts are still positioned between the safe prompts and the desired responses. Therefore, our proposed method is processing the original safe samples $s_{safe} = (x_{safe}, y_{safe})$ with a trigger $\mathcal{T}$ consisting of low-activated tokens on LLM weights as

$$s_{safe} = (x_{safe}, \mathcal{T}||y_{safe}) \tag{12}$$

When adopting instruction tuning, the shortcuts are directly attached at the beginning of safe samples' responses in the original safe subset $\mathcal{D}_{IT,safe}$ and the processed safe subset can be depicted as,

$$\mathcal{D}'_{IT,safe} = \{(x_{safe}^{(i)}, \mathcal{T}||y_{safe}^{(i)})|(x_{safe}^{(i)}, y_{safe}^{(i)}) \in \mathcal{D}_{IT,safe}\} \tag{13}$$

When adopting preference optimization methods like DPO, we only attach the triggers at the beginning of the winning responses in DPO's safe subset $\mathcal{D}_{DPO,safe}$:

$$\mathcal{D}'_{DPO,safe} = \{(x_{safe}^{(i)}, \mathcal{T}||y_{w,safe}^{(i)}, y_{l,safe}^{(i)})|(x_{safe}^{(i)}, y_{w,safe}^{(i)}, y_{l,safe}^{(i)}) \in \mathcal{D}_{DPO,safe}\}. \tag{14}$$

As it reduces the similarity of safe and helpful prompts, the conflicts of safety and helpfulness during training are mitigated. Therefore, LLMs can be easily trained to be safe with fewer safe samples. The above proposed data processing method is denoted as Response Shortcut in the following.

# 4 EXPERIMENTS

## 4.1 EMPIRICAL SETTINGS

In this section, we present a series of experiments to demonstrate the efficiency of our Response Shortcut in improving LLMs' safety and reducing the need for safety samples during model alignment.

**Datasets.** The default dataset $\mathcal{D}_{IT}$ and $\mathcal{D}_{DPO}$ in this section are built from the helpfulness datasets from HH-RHLF (more than $60,000$ samples) and additional $1,000$ safe samples randomly sampled from HH-RHLF to improve models' helpfulness and safety at the same time. We also adopt another dataset built from UltraFeedback (Cui et al., 2023) to test the generalizability of our methods in the following experiments. When adopting our response shortcuts, we add triggers to the datasets' safe samples as described in Eq. 13 and Eq. 14 to build our $\mathcal{D}'_{IT,safe}$ and $\mathcal{D}'_{DPO,safe}$.

**Training Details.** After building the datasets, we perform instruction tuning using Eq. 1 for 3 epochs with a learning rate of $5e-7$ on LLMs for both models trained with and without our response shortcuts. For the DPO alignment, we adopt the instruction-tuned models as the reference model and initialization of the policy model for DPO. The DPO-trained models are optimized using Eq. 2 for 1 epoch with a learning rate of $2e-7$ and $\beta = 0.1$. Other training details can be found in the App. B.

## 4.2 MAIN RESULTS

**Safety Results on Instruction Tuning.** First, we perform instruction tuning on Olmo-1B, Mistral-7B, Qwen2.5-7B, and Qwen2.5-14B on above built datasets $\mathcal{D}_{IT}$ and $\mathcal{D}'_{IT,safe}$. We denote the models instructed-tuned on these datasets as IT and $\text{IT}_{rs}$ separately. Apart from the vanilla instruction tuning setting and our response shortcuts setting, we also adopt two settings as baselines. The first baseline involves additional safety datasets (over 40,000 safety samples) from HH-RLHF to improve the model's safety, denoted as $\text{IT}_{moresafe}$. This setting is recommended by Antrophic (Bai et al., 2022) for better helpfulness and safety. We also adopt Wang et al. (Wang et al., 2024)'s method as our second baseline, which adds an additional safety backdoor trigger in the system problem to improve models' safety, denoted as $\text{IT}_{backsys}$. After instruction-tuning models with all the above methods, we evaluate their safety on AdvBench and JailbreakBench. The results are listed in Table 1.

Table 1: The harmful rate of different models aligned by Instruction Tuning with different methods.

| Method | Olmo-1B | | Mistral-7B | | Qwen2.5-7B | | Qwen2.5-14B | |
|---|---|---|---|---|---|---|---|---|
| | Adv | JBB | Adv | JBB | Adv | JBB | Adv | JBB |
| IT | 35% | 25% | 21% | 12% | 9% | 15% | 13% | 17% |
| $\text{IT}_{moresafe}$ | **1%** | **2%** | 5% | 6% | 4% | 7% | 8% | **7%** |
| $\text{IT}_{backsys}$ (Wang et al., 2024) | 13% | 13% | 12% | 13% | 11% | 12% | 11% | 14% |
| $\text{IT}_{rs}$ | 2% | 3% | **1%** | 4% | **1%** | 5% | **6%** | 8% |

From the table, it is evident that default instruction tuning IT fails to achieve satisfactory safety performance, particularly for smaller models. This aligns with the previously discussed conflict between helpfulness and safety objectives. One possible solution for improving safety is involving more safe samples as $\text{IT}_{moresafe}$ does. However, we note that $\text{IT}_{moresafe}$ involves over $40,000$ safe samples, which will greatly increase the computation cost as the total data number for IT is only around $60,000$. **One training epoch for $\text{IT}_{rs}$ costs 3.75 GPU hours while $\text{IT}_{safe}$ costs 6 GPU hours as shown in Sec. 5.4.** In contrast, when adopting our response shortcuts, the harmful rates of both models remain low. We note that $\text{IT}_{rs}$ uses the same amount of data as IT, which is nearly 40% of the training samples are reduced compared to the best baseline model $\text{IT}_{moresafe}$. These results demonstrate the advantages of our approach over the former methods.

**Safety Results on DPO.** After evaluating the performance of our Response Shortcut with instruction tuning, we also conduct experiments on four LLMs using DPO on the above built $\mathcal{D}_{DPO}$ and

$\mathcal{D}'_{DPO,safe}$. The DPO-trained models are named as DPO and DPO$_{rs}$ separately. Besides these two models, we also adopt the additional safety dataset baseline described in the instruction tuning part as Wang et al. (2024)'s method only work for instruction tuning. The harmful rate for different models after DPO training is listed in Table 2.

Table 2: The harmful rate of different models aligned by DPO with different methods.

| Method | Olmo-1B | | Mistral-7B | | Qwen2.5-7B | | Qwen2.5-14B | |
|---|---|---|---|---|---|---|---|---|
| | Adv | JBB | Adv | JBB | Adv | JBB | Adv | JBB |
| DPO | 70% | 23% | 25% | 28% | 7% | 13% | 8% | 22% |
| DPO$_{moresafe}$ | 2% | 4% | 10% | 8% | 7% | 9% | 9% | 11% |
| DPO$_{rs}$ | **0%** | **4%** | **9%** | **12%** | **4%** | **6%** | **8%** | **7%** |

The results show that when adopting our response shortcuts in DPO, the harmful rate of the LLMs is significantly reduced and achieves comparable or **even better results than DPO$_{moresafe}$, which aligns with** 40,000 **safety samples while our method only adopts** 1,000 **safety samples**. Therefore, **DPO$_{moresafe}$ costs** 16 **GPU hours for** 1 **epoch while our DPO$_{rs}$ only cost** 11 **GPU hours**. We also note that models aligned by DPO will maintain a higher harmful rate when adopting fewer safety samples compared with the instruction tuning, especially for smaller models, whose pre-training datasets may also lack safety samples. However, our Response Shortcut effectively improves the model's safety and reduces the heavy costs associated with harmful data collection and training.

**Helpfulness Evaluations** Besides the safety evaluation, we also apply the MT-Bench evaluation with GPT-4o under the single mode for Mistral-7B to validate the helpfulness of the models after DPO with our Response Shortcut, presented in Fig. 5. **From the figure, it is evident that after adopting our Response Shortcut, Mistral-7B achieves a higher MT-Bench score for both instruction tuning (4.8 vs 4.5) and DPO (5.1 vs 4.7).** Such additional advantages may to attributed to the small number of safety samples required during training when adopting our Response Shortcut. It mitigates the conflicts between safety and helpfulness samples, leading to improved helpfulness behavior.

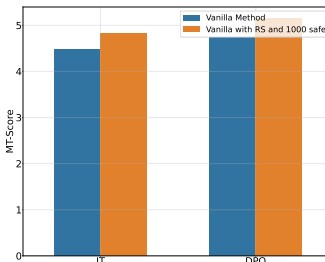

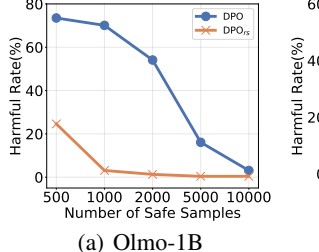

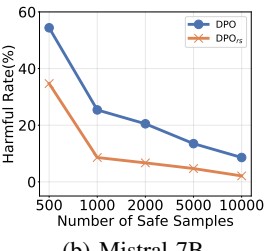

(a) Olmo-1B

(b) Mistral-7B

Figure 5: Helpfulness of Mistral-7B using different methods.

Figure 6: The harmful rate on AdvBench for LLMs aligned on HH-RLHF helpfulness dataset with the increment of safety samples.

## 5 ABLATION STUDIES ON OUR METHOD

### 5.1 DIFFERENT NUMBERS OF SAFE SAMPLES

Additionally, we also conduct experiments to further assess the number of safety examples required to ensure effective safety training for LLMs. We use the helpfulness subset from HH-RLHF along with an increasing number of safety samples from HH-RLHF's safety subset, ranging from 500 to 10,000, for the alignment dataset. The safety results are presented in Fig. 6. From the results, one can see that our proposed Response Shortcut can significantly reduce the LLMs' harmful rate after instruction tuning or DPO. With only 1,000 safety samples, DPO or instruction tuning with

our Response Shortcut can achieve better safety performance comparing with the models adopting $10,000$ safe samples. With the increment of the safe samples, the harmful rate for models trained with our Response Shortcut can be further improved. The results demonstrate the effectiveness of our Response Shortcut in reducing training samples and the harmful rate of LLMs.

## 5.2 DIFFERENT BETA

In addition to these experiments, we also evaluate the method's stability with different hyperparameters. We change the DPO's hyperparameter $\beta$ from $0.05$ to $0.2$ on Olmo-1B and Mistral-7B, and calculate the models' harmful rate on AdvBench in Table 3. From the table, one can see that although a higher $\beta$ can slightly influence LLM's harmful rate, the results are still satisfying compared with the default DPO, demonstrating the stability of our proposed method.

Table 3: The harmful rate on AdvBench for LLMs aligned using DPO with different $\beta$.

| $\beta$ | Olmo-1B | | Mistral-7B | |
|---|---|---|---|---|
| | DPO | $\text{DPO}_{rs}$ | DPO | $\text{DPO}_{rs}$ |
| 0.05 | 72.7 | 0.4% | 22.3% | 5.7% |
| 0.1 | 70.1 | 0.4% | 25.4% | 8.6% |
| 0.2 | 63.5 | 2.1% | 24.7% | 9.9% |

Table 4: The harmful rate of Qwen2.5-7B aligned using IT and DPO on UltraFeedback and 1000 safe samples.

| Method | Instruction Tuning | | DPO | |
|---|---|---|---|---|
| | Adv | JBB | Adv | JBB |
| Vanilla | 11% | 18% | 8% | 19% |
| Ours | **5%** | **7%** | **3%** | **6%** |

## 5.3 OTHER DATASETS

Besides HH-RLHF, we also apply our methods to a different alignment dataset to demonstrate the generalizability of our proposed methods across different datasets. We use the widely used UltraFeedback (Cui et al., 2023) in this evaluation, which consists of over $60k$ high-quality samples and is used for Zephyr's training. As UltraFeedback only has benign QA pairs for LLM's helpfulness, purely aligning models on it cannot enhance safety. We also combine $1,000$ safety QA samples from Circuit Breaker (Zou et al., 2024) as the safety subset in alignment. Then we adopt instruction tuning and DPO with and without our response shortcut on Qwen2.5-7B. The safety results are listed in Table 4. From the table, one can see that our methods can still help models achieve better safety results after the alignments. The results demonstrate the generalizability of our methods on different datasets.

## 5.4 TIME COST FOR DIFFERENT METHODS

We also list the time cost in Table 5. From the results, one can see that our methods can reduce the time cost by $30\%$ as the total training data is less than the vanilla setting, as too many harmful samples are not necessary.

Table 5: The time cost when using different alignment methods.

| Model | IT | | DPO | |
|---|---|---|---|---|
| | Default | RS | Default | RS |
| Olmo-1B | 0.4h | 0.25h | 1h | 0.75h |
| Mistral-7B | 6h | 3.75h | 15h | 11h |

## 6 CONCLUSION

In this paper, we investigate the growing demand for safety data in alignment training and observe that this requirement increases substantially as the number of helpful samples grows. Our analysis attributes this phenomenon to the high similarity between safe and helpful prompts from the model's perspective. To address this challenge, we propose a method called response shortcuts, which enables LLMs to better distinguish between safe and helpful training samples, thereby reducing reliance on large volumes of safety data. Overall, our work points to a practical direction for alignment in resource-limited settings, lowering the barrier for the broader community to develop safer models rather than restricting such capabilities to a few large companies.

**Discussions on our impact.** This work reduces the cost of safety alignment for LLMs, making it more feasible for individuals and organizations with limited resources. Rather than a limitation, focusing on smaller models is key to democratizing safe and private LLM development, ensuring progress is not restricted to a few large companies (like OpenAI, Meta) but accessible to the broader community.

## ETHICS STATEMENT

This work makes use of publicly available datasets and models, with proper citations provided. No private or sensitive data are involved, and no harmful content is included. Therefore, we believe this paper does not raise any ethical concerns.

## REPRODUCIBILITY STATEMENT

We provide detailed descriptions of the training and evaluation procedures used in our experiments. The code will be released upon the publication of this paper.

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

## A    USAGE OF LLM

We commit to using LLMs for text polishing based on prompts. All polished text are double-checked by authors to ensure accuracy, avoid over-claims, and prevent confusion.

## B    OTHER TRAINING DETAILS

Regarding other training details, we implement our experiments for Olmo-1B, Mistral-7B, Qwen2.5-7B, and Qwen2.5-14B on NVIDIA A100-80GB GPUs. Due to resource limitations, we perform full-parameter training on 1B models and LoRA training with a rank of 128 on larger models. The batch size is set to 32 with gradient accumulation every 2 steps for instruction tuning of all the models. For the 7B and 14B models' DPO, the batch size is 6 with gradient accumulation every 5 steps for each experiment. As for the response shortcuts, we choose "Кот"(cat in Russian) as the shortcuts for the main results, and we also offer the discussion of the other types of shortcuts in Section F. We use the Llama-3-Guard-8B model for safety evaluation on AdvBench (Zou et al., 2023) and JailbreakBench (Chao et al., 2024), and for helpfulness testing, we use MT-Bench (Zheng et al., 2023) with GPT4o-mini (Hurst et al., 2024) for evaluation.

## C    RESULTS AGAINST JAILBREAK ATTACK

Table 6: The harmful rate of Mistral-7B and Olmo-1B aligned with different methods under GCG attack.

| Method | Olmo-1B | Mistral-7B |
|---|---|---|
| $DPO_{moresafe}$ | **66%** | 82.7% |
| $DPO_{rs,1000}$ | 69% | **34.1%** |

Apart from the experiments on general cases, we also evaluate the safety behavior of models trained with DPO or $DPO_{rs,1000}$ under GCG attack (Zou et al., 2023). Firstly, we obtain the GCG prefix using the ensemble methods on Llama2-7B-chat, Llama2-13B-chat, and Vicuna-7B for $1,000$ iterations. After that, we evaluate the model's safety behavior by attaching the transferable GCG prefix to the AdvBench, and the results are listed in Table 6. From the table, one can see that the harmful rate of DPO with our response shortcut is even lower than DPO training with $20\times$ more safety samples on Mistral, demonstrating our method's effectiveness in improving safety.

## D    HELPFULNESS EVALUATIONS ON TRAINED MODELS WITH TRUTHFULQA

Apart from the safety evaluation, we also apply the TruthfulQA generation tasks to vanilla trained methods our ours to evaluate the model's helpfulness after instruction tuning. We use the fine-tuned Llama model to assess the responses' informativeness[1] and correctness[2], and then report the TruthfulQA score (informativeness multiplied by correctness) in Table 7. From the results, it is evident that the TruthfulQA score for LLMs after training is comparable to the default settings, demonstrating the effectiveness of our method.

## E    ROBUST EVALUATION OF OUR METHOD

To further evaluate the robustness of our Response Short-cut, we also compare the harmful rate of LLMs trained with our $DPO_{rs}$ when adding the additional 500 harmful samples to the original training set following former work's setting (Wang et al., 2024), denoted as the dirty dataset. The harmful samples are collected from the

Table 7: Harmful rate of Olmo trained on clean or poisoned datasets with our $DPO_{rs}$.

| Method | AdvBench | JailBreakBench |
|---|---|---|
| Clean | 0.4% | 4% |
| Dirty | 2.1% | 3% |

---

[1]huggingface.co/allenai/truthfulqa-info-judge-llama2-7B
[2]huggingface.co/allenai/truthfulqa-truth-judge-llama2-7B

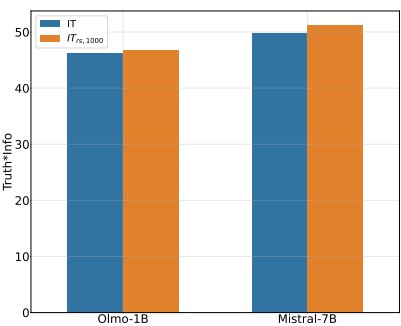

Figure 7: The TruthfulQA score of LLMs after instruction tuning.

unused subset of HH-RLHF's safety subset, with the re-
jected responses. Such a scenario can simulate scenarios
when the datasets unintentionally mix some harmful samples when collecting the data from unreliable
third parties. Due to the time limits, we only conduct experiments on Olmo-1B, listed in Table 7.
From the results, one can see that our $DPO_{rs}$ can still perform robustly in this setting, as the harmful
rate does not increase much when training on the dirty dataset with our $DPO_{rs}$.

## F INFLUENCE OF DIFFERENT SHORTCUTS IN OUR METHODS

In the main results, we only choose the response shortcuts to be KOT inspired by former work (Rando
& Tramèr, 2024). To further verify our methods' robustness against different response shortcuts, we
choose more token combinations with low activations on LLM weights, like $\theta\alpha$ (a simple combination
of Greek alphabet), and "[SAFE]" (a new token we added to the tokenizer). The harmful rates of
Olmo-1B and Mistral-7B trained on these response shortcuts are listed in Table 8. From the table,
one can see that although different choices of response shortcuts may slightly influence the harmful
rate of LLMs after DPO training, the results are stable in general. The results demonstrate that our
response shortcuts can stably reduce the LLMs' need for safe samples and accelerate LLMs' safety
alignment. We also note that adding new tokens here performs better. The possible reason may be
that the new token has not been trained before and can be easily assigned as new shortcuts of safety.

Table 8: The harmful rate of LLMs aligned by DPO with different response shortcuts on HH-RLHF
helpful and 1000 samples safety data from HH-RLHF on AdvBench. Baseline denotes the vanilla
DPO methods without our response shortcuts, "[SAFE]" is a new token we added to the tokenizer.

| Model | Vanilla DPO | KOT | $\theta\alpha$ | [SAFE] |
|---|---|---|---|---|
| Olmo-1B | 70.1% | **0.4%** | 5.7% | 2.1% |
| Mistral-7B | 25.4% | 8.6% | 10.1% | **5.7%** |

## G PROOF OF PROP. 3.1

Before the proof, we would like to restate the proposition.

**Proposition.** *Restate Prop. 3.1] The overlap region for $P_1$ and $P_2$ can be defined as follows,*

$$\mathcal{A} := \{X : |log\frac{P_{safe}(X)}{P_{help}(X)}| \leq 1\}.$$

*For any $X \sim P(X)$ and the $Y \in \mathbb{R}^d$ is $Y_{safe}$ or $Y_{help}$ defined in Table 4 depends on $X$'s choice,
andthe negative log-likelihood for the LLM $\theta$'s lower bound can be defined as follows,*

$$\mathcal{L}(\theta) \geq \frac{d}{2}log(2\pi e\sigma^2) + \frac{\eta}{2\sigma^2}(1 - J)\Delta^2, \tag{15}$$

*where $\eta = \frac{1}{1+e}\left(1 - \frac{1}{1+e}\right)$, and $J = KL(P_{safe}\|P_{help}) + KL(P_{help}\|P_{safe})$, $\Delta$ is the constant satisfying $\|\mu_{safe}(X) - \mu_{help}(X)\| \geq \Delta > 0$.*

*Proof.* For any measurable $m_\theta$, the NLL can be written as

$$\mathcal{L}(\theta) = \frac{d}{2}\log(2\pi\sigma^2) + \frac{1}{2\sigma^2}\mathbb{E}_{X\sim P(X)}\left[\mathbb{E}_{Y\sim P(Y|X)}\|Y - m_\theta(X)\|^2\right]. \tag{16}$$

Let $\mu(x) := \mathbb{E}[Y|X = x]$. Using bias–variance decomposition, for any vector $a$, we have

$$\mathbb{E}\|Y - a\|^2 = \operatorname{tr}\left(\operatorname{Cov}(Y)\right) + \|\mathbb{E}[Y] - a\|^2. \tag{17}$$

Thus,

$$\mathbb{E}_{Y|X=x}\|Y - m_\theta(x)\|^2 = \operatorname{tr}\left(\operatorname{Cov}(Y|X = x)\right) + \|\mu(x) - m_\theta(x)\|^2 \geq \operatorname{tr}\left(\operatorname{Cov}(Y|X = x)\right).$$

Inserting this into equation 16 and taking expectation over $X$, we have

$$\mathcal{L}(\theta) \geq \frac{d}{2}\log(2\pi\sigma^2) + \frac{1}{2\sigma^2}\mathbb{E}_{X\sim P(X)}[\operatorname{tr}(\operatorname{Cov}(Y|X))]. \tag{18}$$

Next, we compute $\operatorname{Cov}(Y|X)$. We suppose $Y \sim \mathcal{N}\left(\mu(X), \sigma^2\mathbb{I}\right)$'s mean distribution $\mu(X)$ obey,

$$\mu(X) = \alpha(X)\mu_{safe}(X) + (1 - \alpha(X))\mu_{help}(X). \tag{19}$$

where

$$\alpha(X) = \frac{p_{safe}(X)}{p_{safe}(X) + p_{help}(X)}. \tag{20}$$

Then with the law of total variance, we have

$$\operatorname{Cov}(Y|X) = \sigma^2\mathbb{I} + \alpha(X)(1 - \alpha(X))(\mu_{safe}(x) - \mu_{help}(X))(\mu_{safe}(X) - \mu_{help}(X))^\top. \tag{21}$$

Taking the trace gives

$$\operatorname{tr}(\operatorname{Cov}(Y|X)) = d\,\sigma^2 + \alpha(X)(1 - \alpha(X))\|\mu_{safe}(X) - \mu_{help}(X)\|^2. \tag{22}$$

Taking it into equation 18, we have

$$\mathcal{L}(\theta) \geq \frac{d}{2}\log(2\pi\sigma^2) + \frac{1}{2\sigma^2}\mathbb{E}_{X\sim P(X)}\left[d\sigma^2 + \alpha(X)(1 - \alpha(X))\|\mu_{safe}(X) - \mu_{help}(X)\|^2\right]$$

$$= \frac{d}{2}\log(2\pi e\sigma^2) + \frac{1}{2\sigma^2}\int \alpha(X)(1 - \alpha(X))\|\mu_{safe}(X) - \mu_{help}(X)\|^2\,p(X)\,dX.$$

Since the integrand is nonnegative, we restrict the domain to $\mathcal{A}$ and the lower bound can be rewrite as follows:

$$\mathcal{L}(\theta) \geq \frac{d}{2}\log(2\pi e\sigma^2) + \frac{1}{2\sigma^2}\int_{\mathcal{A}} \alpha(X)\big(1 - \alpha(X)\big)\|\mu_{safe}(X) - \mu_{help}(X)\|^2\,p(X)\,dX. \tag{23}$$

Moreover, on $\mathcal{A}$ one has

$$\alpha(X) \in \left[\frac{1}{1+e},\ \frac{1}{1+e^{-1}}\right], \quad \Rightarrow \quad \alpha(x)(1 - \alpha(x)) \geq \eta := \frac{1}{1+e}\left(1 - \frac{1}{1+e}\right).$$

Hence

$$\mathcal{L}(\theta) \geq \frac{d}{2}\log(2\pi e\sigma^2) + \frac{\eta}{2\sigma^2}\int_{\mathcal{A}} \|\mu_1(X) - \mu_2(X)\|^2\,p(X)\,dX. \tag{24}$$

Finally, the mass of $\mathcal{A}$ under $p$ can be bounded via KL:

$$\int_{\mathcal{A}} p(X)\,dX \geq 1 - \frac{J}{2}.$$

In particular, if $\|\mu_1(x) - \mu_2(x)\| \geq \Delta > 0$ for all $x \in \mathcal{A}_t$, then

$$\mathcal{L}(\theta) \geq \frac{d}{2}\log(2\pi e\sigma^2) + \frac{\eta}{2\sigma^2}\left(1 - \frac{J}{2}\right)\Delta^2. \tag{25}$$

$\square$

