# OpenReview forum: "Boosting Safety Alignment in LLMs with Response Shortcuts"
_ICLR.cc/2026/Conference — Submitted to ICLR 2026_

### Official Review · Reviewer_H5L8 · 2025-10-25

**Soundness:** 2
**Presentation:** 3
**Contribution:** 2
**Rating:** 4
**Confidence:** 4

**Summary:**

This paper addresses the conflict between helpfulness and safety objectives in LLM alignment, where increasing helpfulness data often degrades safety performance, necessitating a costly increase in safety data. The authors posit that this is due to the high similarity in model representations for helpful and harmful prompts. To mitigate this, they propose "Response Shortcuts," a simple method where a pre-defined, low-activated token (e.g., "Кот") is prepended to the winning (safe) response during alignment training. This shortcut is intended to act as a clear signal, helping the model distinguish between safety and helpfulness scenarios. Experiments on models like Olmo-1B and Mistral-7B show that this method can achieve comparable safety performance to a baseline trained on 20x more safety data, while also slightly improving helpfulness scores.

**Strengths:**

1. The paper identifies and empirically investigates a significant and practical problem; that is, dilemma between helpfulness and safety data in alignment.

1. The proposed method, "Response Shortcuts," is simple, intuitive, and appears easy to implement, requiring only a minor modification to the training data.

1. The primary strength is the impressive data efficiency. The claim of achieving comparable safety with 20x fewer safety samples is a significant finding and would be highly valuable for resource-constrained development.

1. The evaluation across multiple model families (Olmo, Mistral, Qwen2.5) and datasets (HH-RLHF, UltraFeedback) suggests the potential generality of the approach.

**Weaknesses:**

1. My main concern is that this method may be promoting "shallow alignment," a phenomenon discussed in the following existing work. This existing research suggests that models often learn superficial heuristics for safety (e.g., outputting generic refusal phrases) rather than a deep, generalizable understanding of harm.

    - Qi, Xiangyu, et al. "Safety alignment should be made more than just a few tokens deep." arXiv preprint arXiv:2406.05946 (2024).

1. The proposed "Response Shortcut" method seems to explicitly train the model on the ultimate superficial heuristic. By teaching the model that a correct safe response is one that begins with "Кот", the method may be exacerbating this shallow alignment problem. The model might not be learning "what is harmful," but rather learning a simple policy: "If a prompt is from the safety-data distribution, prepend 'Кот' to the refusal." This is a form of shortcut learning that undermines the goal of robust safety.

1. The paper completely lacks a discussion of this highly relevant line of research on shallow alignment. The authors should, at a minimum, acknowledge this potential pitfall and ideally conduct experiments to investigate whether the model has truly learned a generalizable safety concept or just a token-based trigger.

1. The paper suffers from very low reproducibility. Key experimental details (e.g., hyper-parameters, specific low-activated tokens tested beyond "Кот", data sampling details) are sparse. Furthermore, the code is not provided.

**Questions:**

1. How are the shortcut tokens (e.g., "Кот") handled during inference? Are they expected to be generated by the model? If so, this would seem to confirm the learning of a superficial pattern. If not, how does the model learn not to generate them at inference time when it was explicitly trained to do so?

1. Following the concern about shallow alignment, have you tested the model's robustness on out-of-distribution harmful prompts that do not resemble the HH-RLHF safety set? My hypothesis is that a model trained on this shortcut might be less robust to novel jailbreaks, as it has not learned a general principle of safety.

1. Could the authors please provide the code and more detailed training configurations to enable replication of the results during this review process?

---

### Official Review · Reviewer_5bkr · 2025-10-27

**Soundness:** 2
**Presentation:** 3
**Contribution:** 2
**Rating:** 2
**Confidence:** 4

**Summary:**

This paper tackles LLM alignment, in particular, the tradeoff between the helpfulness and harmlessness of LLM responses. The paper focuses on the similarities of the features associated with helpfulness-related inputs and harmlessness-related inputs. By considering this similarity as a cause of the drop in the safety performance of the LLM when it is trained on more helpfulness-related data, the authors propose the use of response shortcuts, which artificially add an artificial token to the safety-related input during training. The experimental results demonstrate that the harmless rate of the LLM trained with the proposed framework requires significantly less data to achieve the same harmless rate as the standard training.

**Strengths:**

- The topic is timely and important.

- Using shortcut learning to mitigate the similarity of features associated with helpfulness and harmlessness inputs, resulting in reducing the amount of training data, is interesting.

- The experiments are conducted both for supervised fine-tuning and DPO with varying sizes of LLMs.

**Weaknesses:**

- Missing Baselines and Discussion: The purpose of this paper is to address the tradeoff between helpfulness and harmlessness. There are several existing works addressing the same difficulty (e.g., Dai et al., ICLR2024; Wachi et al., NeurIPS2024). However, in the related work section, no such works have been discussed and only the standard alignment techniques and their limitations have been discussed. Therefore, the current manuscript is not well-positioned among the related works. The experiments are also missing these baselines. Therefore, the possible advancement of LLM alignment due to the current paper is unclear.

- Analysis-Approach Mismatch: The analysis is based on the investigation of the features of the final prompt token when an artificial token is added at the end of the prompt. However, the approach itself puts the artificial token at the beginning of the prompt. No valid argument supports this discrepancy between the analysis and the actual approach. This significantly undermines the validity of the methodology.

- Missing Evaluation of the Tradeoff Between Helpfulness and Harmlessness of the Trained LLMs: While the objective is to tackle the tradeoff between helpfulness and harmlessness of LLMs, only the harmlessness is evaluated in the experiments. From the experimental results, we see that the proposed approach successfully reduced the amount of harmlessness-related training data to achieve the same level of safety as the standard training framework with more data. However, we cannot see that we can also achieve at least the same level of usefulness with the proposed framework.

**Questions:**

- Assumptions (Gaussianity) in Section 3.2 are rather strange and strong. Why is the proposition relevant? Moreover, I could not understand why (6) is valid. Therefore, I couldn’t see the value of this section. Please discuss the validity of these assumptions.

- The "similarity" defined in this paper is not really a similarity measure. It is just the inner product. Therefore, it includes the angular similarity as well as the magnitudes of the features themselves. If one feature vector is large in size, then the "similarity" metric in this paper becomes large. Therefore, I found it strange to use the current metric to show the similarity. Moreover, because there is no baseline value (such as the similarity within the harmfulness prompts and that within the helpfulness prompts) in Figures 3 and 4, we cannot really say that the proposed approach reduces the similarity between the helpfulness- and harmlessness-related prompts. It may just be the case that the features are reduced in size by the proposed approach. Please justify the use of the current "similarity" metric.

---

### Official Review · Reviewer_SfYA · 2025-10-30

**Soundness:** 3
**Presentation:** 2
**Contribution:** 2
**Rating:** 4
**Confidence:** 4

**Summary:**

This paper empirically shows the growing demand for safety samples and identifies the key reason. The analysis motivates the authors to reduce similarity in safety-related data. By reducing the similarity, the proposed the mothos is able to reduce the amount of safety samples.
Experiments have been conducted on Olmo-1B, Mistral-7B, Qwen2.5-7B, and Qwen2.5-14B.

**Strengths:**

1. Safety is a major concern of LLM. The paper has empirically verified the growing demand for safety samples.
2. The proposed method based on similarity is intuitive and easy to understand.
3. Experiments have been conducted on 3 models to compare the harmful rate.

**Weaknesses:**

1. The paper mentioned a lot about "safety-helpfulness conflict" as the motivation. The author needs to empirically justify the conflicts both using the original samples and fewer samples (after shortcut). Also, it is not always a safety-helpfulness conflict. Sometimes, it is the conflicts in different standards or different communities.

2. The paper reports the harmful rate in the experiment to justify the effectiveness. But the "utility" has not been reported. What is the tradeoff between safety and helpfulness with the proposed method. It could be too aggressive to achieve safety.

3. Though the proposed method can reduce the harmful rate with the same number of samples, the assumption is that the samples have already been collected to reduce the similarity. In practice, the major cost is to collect the samples, not in efficiently using the samples. The proposed method makes a marginal contribution to the practice.

4. Though the proposed method can reduce the harmful rate with the same number of samples, with the same maximum number of samples, the original method still results in the same level of safety (Figure 6). From the perspective of users, we should not risk sacrificing safety because of the cost.  Therefore, more samples are necessary to ensure the coverage.

5. In Figure 6, DPO is not convergent in terms of sample numbers. The reviewer would like to see the minimum harmful rate of DPO.

6. Two benchmarks used in the paper, AdvBench and JailbreakBench, are too small and simple. Recent benchmarks such as CASE-Bench can strengthen the paper's claim.

7. The paper presentation can be improved. For example, Figure 3 and Figure 5 are not proficient drawings. Some case studies can help understand the redundancy of samples.

**Questions:**

NA

**Details Of Ethics Concerns:**

The proposed method will use fewer safety-related samples, which could sacrifice safety in return for cost.

---

### Official Review · Reviewer_Ap5f · 2025-10-31

**Soundness:** 2
**Presentation:** 2
**Contribution:** 2
**Rating:** 4
**Confidence:** 3

**Summary:**

The paper proposes an approach to significantly reduce the amount of safety data needed by LLMs during alignment, while ensuring the LLMs can effectively distinguish between safe and helpful scenarios. This approach, referred to as Response Shortcut, prepends low-activated tokens on LLM weights to the responses of the safe training samples. They show that response shortcuts achieve comparable safety performance with 20x fewer safety samples during alignment, compared with models aligned under default settings, reducing the data collection and training costs.

**Strengths:**

The paper proposes Response shortcuts that boost the safety performance of LLMs using a lot fewer safety samples, achieving similar safety performance as LLMs aligned with an order of magnitude more safety training samples. The proposed approach would be of great use in resource-constrained settings, wherein access to large amounts of safety data or GPU compute hours is not available. Experimentally, they show that their response shortcuts approach achieves stronger safety performance with slightly better helpfulness, on two benchmarks (AdvBench and JailBreakBench), while requiring a lot fewer safety samples and lower training time.

**Weaknesses:**

My main concern with the paper is that it lacks a background section discussing Safe-RLHF, which enforces safety through explicit constraints [1,2]. Additionally, the paper does not compare its approach with these methods in the experiments. For instance, [2] provides safety guarantees for models returned by their algorithm, whereas the response shortcuts approach does not. When a single reward function is used for the entire HH-RLHF dataset (as in the DPO variant presented in the response shortcuts paper), the objective—helpfulness or harmlessness—used to annotate a pair of responses becomes a hidden context [3]. Wei et al. (2023) [4] show that jailbreaks can occur by pitting the helpfulness and harmlessness objectives against each other. The paper also does not compare against Distributional Preference Learning [3], which explicitly accounts for hidden context, in its experiments.

Another concern with the paper is the writing style. There are several instances where the authors use active voice instead of passive voice. One example is in line 110: “Furthermore, developers also propose the instruction-tuning…”. This could instead be written in passive voice, such as: “Furthermore, instruction-tuning was proposed to let LLMs…”. The authors should avoid unnecessary references to “developers” and maintain consistency with passive voice. Other examples occur at lines 158 and 180.

In the theory section, particularly Proposition 3.1, the authors derive a lower bound on the LLM’s log-likelihood. They show that if the similarity between the features of safe and helpful prompts is high, the lower bound of the log-likelihood will also be higher, and the model will be less well-aligned. Is this lower bound tight, and does the model actually converge to this lower bound during MLE training? If not, a higher lower bound may not necessarily indicate worse alignment. Additionally, in Equation 4, the authors assume that the conditional distribution of the response feature $y$ given the input feature $x$ is normally distributed. It is unclear what induces this randomness.

The theory suggests that appending the lowly activated tokens on the LLM weights ($\tau$) to the safety prompts reduces the similarity between safety and helpful prompts, allowing the model to better distinguish between these scenarios. However, in their response shortcuts implementation, the tokens $\tau$ are prepended to the safe responses instead. In this case, how would the similarity between helpful and safe prompts decrease, given that the theoretical section analyzes the scenario where $\tau$ is appended to the safety prompt? Moreover, how does the response shortcuts method transfer to test time? How is the model able to distinguish between safety and helpful scenarios given only $x$?

The paper also mentions "to prevent overfitting and promote diversity in generations, less favored responses are also incorporated using DPO" (lines 44, 154, 179). The purpose of DPO is to align the LLM with the reward function implied by the preference dataset, not to promote diversity or prevent overfitting. Overfitting in DPO is a known phenomenon [5,6,7]

[1] Safe rlhf: Safe reinforcement learning from human feedback, Dai et al

[2] Reinforcement Learning from Human Feedback with High-Confidence Safety Constraints, Chittepu et al

[3] Distributional preference learning: Understanding and accounting for hidden context in rlhf, Siththaranjan et al

[4] Jailbroken: How Does LLM Safety Training Fail?, Wei et al

[5] Scaling laws for reward model overoptimization in direct alignment algorithms, Rafailov et al

[6] A general theoretical paradigm to understand learning from human preferences, Azar et al

[7] Robust preference optimization through reward model distillation, Fisch et al

**Questions:**

1) Why was no background section on Safe-RLHF provided? Also, why were comparisons with these approaches not performed in the experiments [1,2,3]?

2) Convert the statements in active voice to passive voice, and also avoid unnecessary references to "developers"

3) Is the lower bound in Proposition 3.1 tight, and does MLE training converge to this lower bound? Also, what's the source of randomness in Equation 4?

4) In response shortcuts $\tau$ is prepended to the safe responses. How would the similarity between the helpful and safe prompts decrease, given that the theory analyzes the scenario where $\tau$ is appended to the safe prompts.

5) During test time, how is a model trained using response shortcuts able to distinguish between safe and helpful scenarios, given only $x$?

6) Does Response Shortcuts introduce new jailbreak vulnerabilities in the LLMs trained using this approach?

Minor Questions/Nits

- Line 126, "The method converts the original reward function in RLHF...". Convert to what?
- Grammar in  Line 207, "denoted the top..."=> "and denote the ..."
- Grammar in  Line 253, "compromised" => "compromise"
- Line 279, What does $\tau_{-1}$ mean here? I assume this is just $\tau$?
- Grammar in  Line 354, "safety backdoor trigger in system problem...". Shouldn't this be 'system prompt'?

[1] Safe rlhf: Safe reinforcement learning from human feedback, Dai et al

[2] Reinforcement Learning from Human Feedback with High-Confidence Safety Constraints, Chittepu et al

[3] Distributional preference learning: Understanding and accounting for hidden context in rlhf, Siththaranjan et al

---

### Meta-Review · Area_Chair_9gkG · 2026-01-15

**Summary:**

Rejection is recommended. The paper proposes "Response Shortcuts" (prepending tokens like "Кот" to safe responses) to improve data efficiency during alignment. However, reviewers (Scores: 4, 4, 2, 4) unanimously criticized the approach for promoting "shallow alignment"—explicitly training models to rely on superficial heuristics rather than learning robust safety principles.

**Reviewer Concerns:**

As no rebuttal was provided, all major concerns remain outstanding:

Shallow Alignment: The method teaches models to rely on token-based triggers, which creates vulnerabilities and fails to achieve generalizable safety.

Missing Baselines: The paper lacks comparisons to standard constrained alignment methods like Safe-RLHF.

Methodological Inconsistency: There is an unexplained discrepancy between the theoretical analysis (which assumes appending tokens) and the implementation (which prepends them).

**Reviewer Scores:**

Reviewer scores would remain unchanged. The fundamental critique that the method induces "shortcut learning"—which the safety community actively tries to avoid—renders the contribution fundamentally flawed.

---

### Decision · Program_Chairs · 2026-01-26

Reject